# Assessment of the TRX2p-yEGFP Biosensor to Monitor the Redox Response of an Industrial Xylose-Fermenting *Saccharomyces cerevisiae* Strain during Propagation and Fermentation

**DOI:** 10.3390/jof9060630

**Published:** 2023-05-30

**Authors:** Raquel Perruca Foncillas, Miguel Sanchis Sebastiá, Ola Wallberg, Magnus Carlquist, Marie F. Gorwa-Grauslund

**Affiliations:** 1Applied Microbiology, Department of Chemistry, Lund University, P.O. Box 124, SE-221 00 Lund, Sweden; raquel.perruca_foncillas@tmb.lth.se (R.P.F.); magnus.carlquist@tmb.lth.se (M.C.); 2Department of Chemical Engineering, Lund University, P.O. Box 124, SE-221 00 Lund, Sweden; miguel.sanchis_sebastia@chemeng.lth.se (M.S.S.); ola.wallberg@chemeng.lth.se (O.W.)

**Keywords:** on-line flow cytometry, redox imbalance, TRX2p-GFP, yeast propagation, lignocellulosic bioethanol, wheat-straw hydrolysate, furfural

## Abstract

The commercial production of bioethanol from lignocellulosic biomass such as wheat straw requires utilizing a microorganism that can withstand all the stressors encountered in the process while fermenting all the sugars in the biomass. Therefore, it is essential to develop tools for monitoring and controlling the cellular fitness during both cell propagation and sugar fermentation to ethanol. In the present study, on-line flow cytometry was adopted to assess the response of the biosensor TRX2p-yEGFP for redox imbalance in an industrial xylose-fermenting strain of *Saccharomyces cerevisiae* during cell propagation and the following fermentation of wheat-straw hydrolysate. Rapid and transient induction of the sensor was recorded upon exposure to furfural and wheat straw hydrolysate containing up to 3.8 g/L furfural. During the fermentation step, the induction rate of the sensor was also found to correlate to the initial ethanol production rate, highlighting the relevance of redox monitoring and the potential of the presented tool to assess the ethanol production rate in hydrolysates. Three different propagation strategies were also compared, and it was confirmed that pre-exposure to hydrolysate during propagation remains the most efficient method for high ethanol productivity in the following wheat-straw hydrolysate fermentations.

## 1. Introduction

Commercially feasible biofuel production from a lignocellulosic biomass requires an optimized process whereby all the sugars present in the raw material are efficiently converted to ethanol. This implies that, independently of the chosen process configuration, the following key steps have to be optimized: pretreatment of the raw material, enzymatic hydrolysis of the pretreated material into fermentable sugars, ethanol fermentation, and product recovery by distillation [1].

The pretreatment of the lignocellulosic raw material aims at increasing the accessibility of the sugar polymers for hydrolysis; during this step, different classes of compounds that inhibit the hydrolysis and fermentation steps can be released: (i) furaldehydes such as furfural and 5-hydroxymethyl furfural (HMF), (ii) phenolic compounds, and (iii) weak acids such as acetic acid [2]. The following steps, hydrolysis and fermentation, can be performed separately or combined in process configurations commonly referred to as separate hydrolysis and fermentation (SHF) and simultaneous saccharification and fermentation (SSF), respectively. When both hexose and pentose sugars can be utilized during SSF, with the help of a xylose-fermenting yeast strain, the process is often referred to as simultaneous saccharification and co-fermentation (SScF). SSF and SScF processes reduce the need for equipment and thus reduce cost by performing simultaneous enzymatic hydrolysis and fermentation in one vessel; furthermore, the SScF process increases the ethanol yield per kilogram of raw material by utilizing both the glucose and xylose fractions. 

Optimally, the fermentation step requires a yeast strain that can consume all the released sugars and has a high tolerance to the released inhibitory compounds. Accordingly, extensive work has been put into obtaining such strains by a combination of metabolic and evolutionary engineering strategies for the generation of robust industrial strains of *S. cerevisiae* [3,4,5]. 

Previous studies have highlighted the relevance of the method used for yeast preparation, commonly referred to as propagation, for achieving an efficient fermentation [6,7,8], as the cellular fitness of the yeast culture at the time of inoculation of the SS(c)F is expected to impact the fermentation efficiency [8]. Different propagation alternatives have been studied, with a strong focus on improving the yeast robustness prior to the SS(c)F step [6,7,9]. In particular, short exposure of the yeast cells to lignocellulosic inhibitors has been shown to improve the following fermentation step by inducing various molecular responses that increase tolerance of the yeast culture [7,9]. More recently, Dobrescu et al. have also proposed the use of a glycerol and ethanol mix as a substrate during propagation to enhance the fermentability of xylose-engineered strains [10]. However, it is still difficult to predict how a new propagation mode, a different propagation duration, a different inhibitor level, or any other change in the propagation parameters can benefit or negatively affect the following fermentation step. This raises the importance of developing in situ systems that can help monitor as well as control the cell fitness during propagation and fermentation.

Among key fitness parameters, the redox balance is known to be affected by the presence of aldehydes due to the need for NAD(P)H cofactor utilization during the yeast-induced reduction into the less toxic alcohol forms [11,12]. Therefore, the use of a transcription factor-based biosensor capable of reporting the redox imbalance in yeast could open the road to a closer monitoring and control of the redox balance during yeast propagation and in the following fermentation step. Such a sensor has been recently designed and optimized by exploiting the native Yap1p response to oxidative stress and utilizing its target promoter TRX2p as the controller for the expression of the fluorescent protein [13,14]. As such, when the cells are exposed to inhibitors known to perturb the redox balance, such as furfural, an activation of TRX2p--and thus an increase of the fluorescent signal of the biosensor--is observed. The fluorescence can be monitored at the population level using fluorometry-based tools. More advanced on-line monitoring at the single cell level can also be obtained by using a flow cytometer combined with automatic sampling [15]. 

In the present study, we explored the application of a TRX2p-yEGFP biosensor-containing yeast with on-line flow -ytometry measurements to monitor the redox state of the cell culture exposed to inhibitory compounds and/or lignocellulosic hydrolysates. This methodology was also used to study three different propagation strategies and their impact on the subsequent fermentation efficiency to ethanol from wheat-straw hydrolysate.

## 2. Materials and Methods

### 2.1. Strains and Media

The industrial *S. cerevisiae* strain cV-110 was provided by Terranol A/S, Copenhagen, Denmark. This strain has previously been engineered by introducing a xylose epimerase, a xylose isomerase (XI), and a xylulokinase (XK) for xylose metabolism and further optimized by extensive laboratory evolution [16,17]. All other yeast strains and shuttle plasmids utilized and developed in this study are listed in Appendix A and Appendix A, respectively. 

For sub-cloning experiments, *Escherichia coli* NEB5α competent cells from New England Biolabs (Ipswich, MA, USA) were also used. Liquid cultures of *E. coli* were performed in a Lysogeny Broth (LB) medium containing 10 g/L tryptone, 5 g/L yeast extract, 5 g/L NaCl, and pH 7.0. Selection of the successful transformants was accomplished in LB agar plates (LB + 15 g/L agar) supplemented with ampicillin (50 mg/L) and incubated overnight at 37 °C. 

For the transformation experiments, the yeast strains were grown in a Yeast Peptone (YP) medium containing 20 g/L peptone and 10 g/L yeast extract and supplemented with 20 g/L glucose or 50 g/L xylose as a carbon source. Shake-flask cultivations were performed at 30 °C and 180 rpm. The selection of transformants was accomplished at 30 °C in YPX agar plates (YP + 50 g/L xylose + 15 g/L agar) supplemented with geneticin (200 mg/L) and nourseothricin (100 mg/L) to select for the Cas9-kanMX and the gRNA-natMX plasmids, respectively.

For the yeast strain characterization, both the rich YP medium and the defined mineral medium [18] were employed and complemented with various carbon sources, as detailed below. 

Wheat straw hydrolysate was also utilized in the propagation and fermentation experiments. This was prepared by impregnating wheat straw provided by Lantbruksprodukter (Lunnarp, Sweden) in H_2_SO_4_ and pretreating it by a continuous steam explosion at 195 °C for 10 min. The impregnation was performed in a dilute sulfuric acid solution for 1 h, and the material was filtered in a filter press (Fischer, Germany) at approximately 13 bar. The soaked fiber was left overnight at room temperature before subjecting it to steam explosion. The concentration of water-insoluble solids (WIS) in the pretreated material was determined following the National Renewable Energy Laboratory (NREL) protocol for determination of insoluble solids in a pretreated biomass [19]. The WIS content of the pretreated material was 12.2%. For experiments where only the liquid fraction was needed, the steam-pretreated material was filtered with a hydraulic filter press, and the liquid fraction, referred to as hydrolysate, was recovered and used. The hydrolysate was also neutralized to pH 5.0 prior to its addition to the culture medium for the small vial fermentations. The composition of the material was analyzed following National Renewable Energy Laboratory NREL protocol for the determination of sugars, by-products, and degradation products in liquid fraction-process samples [20]. The composition can be found in Table 1. 

### 2.2. Construction of Strain TMBRP011

The industrial strain cv-110 was first transformed with the plasmid pCfB2312 using the high-efficiency LiAc protocol [21]. Subsequently, the strain TMBRP011 carrying the biosensor pTRX2_5xUAS-yEGFP was generated the CRISPR/Cas9 system developed by Jessop-Fabre et al. [22], employing the NotI-linearized pRP010 plasmid as the donor DNA. This plasmid was obtained by amplifying the engineered promoter pTRX2_5xUAS from pCfB3236 utilizing primers *TRX2p_f_PstI* and *TRX2p_r_XhoI* and cloning it into pRP005 employing PstI/XhoI restriction sites. The integration was performed in the intergenic region XI-3 (Chr XI: 93378.94567). An extensive screening of the clones was necessary due to the low frequency of the positive clones, with only 4 out of the 44 tested clones having integrated the construct. The verification of the transformants was accomplished by colony PCR in two steps. First, primers *XI-3_ver_f* and *XI-3_ver_r* were utilizedto amplify the integration site XI-3 (Appendix A). Clones that showed a unique band corresponding to the expected size if the integration was successful were further verified by using the specific primer *seqTRX2p* together with *XI-3_ver_r* (Appendix A). 

### 2.3. Shake-Flask Cultivations

The strain TMBRP011 was cultivated overnight in a 50 mL Falcon tube containing 5 mL YPD medium. The cells were recovered and used for inoculation (optical density of 0.2) of 250 mL shake flasks containing 25 mL YPD medium supplemented with 0, 1.25, 2.5 or 5 g/L of furfural. Cultivations were performed at 30 °C and 180 rpm. Samples were collected for optical density (OD_620nm_) and flow cytometry measurements, as detailed below. 

### 2.4. Anaerobic Continuous Cultivations

A two-step pre-cultivation scheme was performed where TMBRP011 was first inoculated in 5 mL YPD medium in a 50 mL Falcon tube for 7 h; the whole culture was used for the inoculation of an overnight pre-culture in a 500 mL baffled shake flask containing 50 mL minimal medium [18] with 20 g/L glucose as the carbon source. The cells were harvested and used as the inoculum for the bioreactor cultivation at an initial OD_620nm_ of 0.3. The continuous cultivation was performed in a 3 L Minifors 2 (Infors HT, Basel, Switzerland) with a 1 L working volume using the same defined mineral medium supplemented with 5 mL/L of antifoam 204, 0.4 g/L Tween-80 and 10 mg/L ergosterol. The temperature was maintained at 30 °C, with the pH at 5.0 by automatic addition of 3 M KOH and stirring at 500 rpm. The bioreactor was continuously sparged with 0.5 L/min N_2_. An initial concentration of 20 g/L of glucose was provided for the batch phase, whereas the feed solution consisted of the same defined medium with 10 g/L glucose. The initiation of the chemostat phase was determined by the drop in CO_2_ production during the batch phase, monitored with a BlueVary gas analyzer (BlueSens, Herten, Germany) for the online detection of exhaust gas. The dilution rate was set to 0.1 h^−1^. A furfural pulse of 2 g/L was added after a steady state was reached (five residence times). Samples were collected for OD_620nm_, HPLC and on-line flow cytometry measurements. 

### 2.5. Aerobic Fed-Batch Yeast Propagation 

TMBRP011 was propagated in a fed-batch mode in 3 L Minifors 2 bioreactors (Infors HT, Basel, Switzerland). Initially, 500 mL of defined mineral medium [18] supplemented with 20 g/L glucose and 20 g/L xylose were inoculated to an initial OD_620nm_ of 0.2. The bioreactor was sparged with 1.2 L/min of sterile air, and the CO_2_ in the exhaust gas was monitored with a BlueVary gas analyzer (BlueSens, Herten, Germany). The feeding was initiated after a drop in the CO_2_ present in the exhaust gas was observed. The feed was started at 10 mL/h for 2 h, then increased to 15 mL/h for another 2 h, and finally increased to 20 mL/h. A total of 500 mL was introduced to reach a final working volume of 1 L. The feed consisted of the same defined mineral medium [18] supplemented with either 20 g/L glucose and 20 g/L xylose (GX propagation), 8.3 g/L glycerol and 33.4 g/L ethanol (GE propagation) or 40% (*v*/*v*) wheat straw hydrolysate (H propagation). In the latter case, sugars were added to complement the feed composition for a final concentration of 20 g/L glucose and 20 g/L xylose. In the GE propagation, the concentration of ethanol was calculated based on a theoretical biomass yield on ethanol of 0.61 g/g [23] to reach a similar biomass as in the GX propagation. Next, the same ratio glycerol/ethanol as used by Dobrescu et al. [10] was utilized to calculate the concentration of glycerol to be added. 

The initial aeration of 1.2 L/min in the batch phase was increased to 1.5 L/min of sterile air in the fed-batch phase. The temperature was maintained at 30 °C, the pH at 5.5 by automatic addition of 3 M KOH and stirring at 600 rpm. Samples were collected for OD_620nm_, HPLC, and on-line flow-cytometry measurements. 

### 2.6. Fermentation on Hydrolysate

After propagation, the equivalent of 3 g cell dry weight (CDW)/L of cells were inoculated into serum vials containing 50 mL defined mineral medium [18] supplemented with wheat-straw hydrolysate. Four different hydrolysate concentrations were tested: 71% (*v*/*v*), 54% (*v*/*v*), 36% (*v*/*v*) or 21% (*v*/*v*) in the serum vials. These volumes corresponded to the inhibitors’ concentrations present in 10% WIS, 7.5% WIS, 5% WIS or 2.5% WIS, respectively. Sugars were complemented accordingly to achieve a final concentration of 20 g/L of glucose and 20 g/L of xylose in all tests. The vials were flushed with N_2_ for at least an hour prior to cultivation and maintained sealed with rubber stoppers. The incubation was performed at 30 °C and 180 rpm. Samples were collected at 0, 1, 3, 5, 8, 24, 48, 72 and 96 h for OD_620nm_, HPLC, and flow cytometry measurements. 

The ethanol yield was calculated based on the available glucose and xylose at the beginning of the fermentation (0 h). 

### 2.7. Simultaneous Saccharification and Co-Fermentation (SScF)

The yeast propagation was performed using the H propagation described above. In this case, the volume of the added feed was increased to 1 L to obtain sufficient biomass for SScF inoculation. The feed composition consisted of 25% wheat-straw hydrolysate (*v*/*v*). 

The whole pretreated slurry was used for SScF experiments carried out in 2 L Labfors bioreactors (Infors HT, Basel, Switzerland) with a total working mass of 1 kg. The SScF was run at a water-insoluble solid (WIS) load of 10%, the enzyme cocktail Cellic CTec2 (Novozymes, Denmark) was added at a load of 0.05 g enzyme/g WIS (corresponding to approximately 10 FPU/g WIS), and 3 g CDW/L of yeast were also added. The medium was supplemented with nutrients consisting of 0.5 g/L (NH_4_)PHO_4_, 0.025 g/L MgSO_4_ and 1 g/L yeast extract. The antibiotics streptomycin and penicillin were added at concentrations of 10 mg/L and 10,000 U/L, respectively, to avoid the risk of infection. Due to the high solid content at the start of the SScF, no mixing was possible until 1 h after adding the enzyme cocktail. At this time, the material had begun liquefying, and mixing at 400 rpm could be started. The SScF was performed at 37 °C, and the pH was maintained at pH 5.0 through the addition of 10% NaOH solution. 

Samples taken after 3 h, 5 h, 8 h, 24 h, 48 h, 72 h and 96 h of inoculation were centrifuged at 13,000 rpm for 5 min. The supernatant was recovered and filtered through 0.2 µm syringe filters for HPLC analysis. 

### 2.8. Analytical Methods

The yeast biomass concentration was estimated by measuring its optical density (OD_620nm_) using an Ultrospec 2100 pro UV/Visible spectrophotometer (Amersham Biosciences, Buckinghamshire, UK). Extracellular metabolites were analyzed using a Waters HPLC system (Milford, CT, USA) equipped with an Aminex HPX-87H column (Bio-Rad, Richmond, VA, USA) operating at 60 °C. The mobile phase was 5 mM sulfuric acid, and the flow rate was maintained at 0.6 mL/min. Glucose, xylose, acetate, furfural, HMF and ethanol were quantified by this method. The correlation between OD_620nm_ and the cell dry weight (CDW) was determined by vacuum filtering 5 mL of culture through pre-weighted filters (0.45 μm pore size membrane; Pall Corporation, New York, NY, USA), followed by washing with MiliQ water and drying for 8 min at 350W in a microwave. The filters were weighed after storage in a desiccator.

### 2.9. Flow Cytometry

Off-line flow cytometry measurements were performed using an Accuri C6+ (BD Biosciences, Franklin Lakes, NJ, USA) equipped with two excitation lasers at 488 nm and 640 nm. The detection filters 533/30 nm, 585/40 nm, 670 LP and 675/25 nm were purchased from BD Biosciences (NJ, USA). When necessary, samples were diluted with phosphate-buffered saline (PBS) at pH 7.4 to OD_620_ < 1.0, stained with 10 µg/mL propidium iodine (PI), and incubated for 10 min. A total of 10,000 events were collected per sample at a flow rate of 35 µL/min. To avoid background noise, a threshold of 80,000 in forward scatter height (FSC-H) was applied. Data analysis was performed using FlowJo^TM^ v10.8.1 software (BD Life Sciences). 

On-line flow cytometry was performed using OnCyt (OnCyt, Switzerland) coupled to an Accuri C6+ (BD Biosciences, NJ, USA). Samples were automatically taken from the bioreactor so that GFP levels and PI staining could be monitored throughout the cultivation. An initial volume of the sample was systematically discarded to prevent measuring the dead volume on the sampling line. Then, the cells were diluted in PBS at pH7.4, stained with propidium iodine (PI), and incubated for 10 min prior to the measurement of the PI staining and GFP levels. Due to the incubation needed for PI staining, a minimal sampling interval of 15 min was used during the furfural pulse, whereas the sampling was performed every hour or three hours afterwards. The samples were run in a fixed-volume mode (35 µL) at a flow rate of 35 µL/min. Thorough cleaning of the instrument was performed when the sampling was performed every hour or three hours. Data were analyzed using FlowJo (BD Life Sciences).

## 3. Results

### 3.1. Evaluation of Sensor Response to Increasing Inducer Concentrations

The TRX2p-yEGFP biosensor for redox imbalance has been shown to induce a response in the presence of furfural in a non-xylose engineered laboratory and industrial strains [14,24]. Here the response of the TRX2p-yEGFP biosensor was evaluated in a robust xylose-consuming industrial strain (TMBRP011, derived from cV-110 [16,17]) by exposing the strain to a range of furfural concentrations (0–5 g/L). The growth in the YPD medium was impacted in all cultures that contained furfural, with no growth detected with 5 g/L furfural and only a slight increase in optical density (OD_620nm_) after 30 h of cultivation with 2.5 g/L furfural (Figure 1A).

No redox response of the biosensor was detected in the absence of furfural, whereas an increase in fluorescence was observed for cultivations where furfural was present, even at the lowest tested concentration (Figure 1B), indicating a similar induction pattern to previously reported strains. Specifically, the culture with the lowest concentration, 1.25 g/L of furfural, was the fastest to show an induction of the biosensor, followed by the 2.5 g/L furfural condition, which eventually gave the highest and most prolonged signal over time. However, the cultivation containing the highest concentration, 5 g/L furfural, showed no induction of the biosensor. This absence of response, together with the lack of growth (Figure 1A), suggests that this concentration of furfural was probably too inhibitory for the cells even to induce the GFP gene. 

Analysis of the fluorescence pattern at the single-cell level indicated that no subpopulations were found (Appendix A), which suggested that the entire population was homogeneously induced. 

### 3.2. Dynamic Response of the Biosensor for Redox Imbalance to Furfural Pulse

To assess the suitability of the biosensor as a reporter of dynamic response, a furfural pulse was performed on the strain TMBRP011, carrying the TRX2p-yEGFP biosensor, which was grown in a chemostat at a dilution rate of 0.1 h^−1^; the fluorescence signal was monitored with on-line flow cytometry to enable a frequency of 15 min between samples throughout the cultivation (>100 h). After the initial batch phase of ca. 22 h, followed by continuous medium feeding for five residence times, 2 g/L of furfural were added as a pulse (Figure 2A). This led to a rapid and significant increase in fluorescence in the GFP channel (Figure 2B), with the highest recorded intensity after 5–7 h, when it peaked and reached a 7.9 ± 1.7 fold-change induction. This was followed by a decrease in fluorescence intensity until uninduced levels were recorded again at ca. 96 h. The putative population heterogeneity was also measured; however, no subpopulation was observed throughout the different stages of the sensor’s response (Figure 2C).

To allow the distinction between intact cells and cells with a permeabilized membrane, propidium iodine (PI) staining was also performed prior to flow-cytometry measurement. Within the six hours after the furfural pulse, no decrease in the percentage of intact cells was recorded (Appendix A), which suggests that although the presence of furfural induced a redox imbalance, it did not harm the integrity of the cell membrane. *S. cerevisiae* has been shown to detoxify furfural by reducing it to its alcohol form [7,12]. This was confirmed here too, as the decrease in furfural levels occurred much faster than the expected rate of furfural disappearance by dilution (Figure 2A). 

The quick and transient redox response observed in the industrial strain confirmed the suitability of the combination of the TRX2p-yEGFP biosensor and an on-line flow cytometry set-up to monitor dynamic changes in cellular redox states during cultivation. 

### 3.3. Effect of Propagation Strategy on the Redox Response and the Fermentation Efficiency of Wheat-Straw Hydrolysate

In the next experiment, the biosensor was used to assess putative correlations between the redox state during the propagation step and the cell performance during the following fermentation step; this was done by monitoring the response of the biosensor using on-line flow cytometry with different propagation media. 

The initial batch phase utilizing a defined mineral medium supplemented with a mixture of 20 g/L of glucose and 20 g/L of xylose as carbon sources was similar for all the propagation strategies. This was followed by three different fed-batch feeding strategies. For the first strategy, the same defined medium containing 20 g/L glucose and 20 g/L xylose was used as feed (GX propagation). For the second strategy, the defined medium was supplemented with 40% (*v*/*v*) wheat-straw hydrolysate (H propagation) and compensated to contain the same amount of glucose and xylose as for the first propagation method. Finally, the third propagation strategy used the non-fermentable carbon sources glycerol and ethanol in the feed (GE propagation), with a similar carbon content as for the first propagation. 

High cell densities were obtained after the propagation in a fed-batch mode for all strategies, albeit with a slightly lower OD for the GE-propagated cells (Figure 3A). The cellular growth was not affected by the level of inhibitors present in the hydrolysate during the H propagation. The integrity of the membrane was not affected either, as no decrease in the percentage of the intact cells was observed during feeding with hydrolysate (Appendix A). 

To assess how the different strategies affected the cellular redox state, the response of the TRX2p-yEGFP biosensor was monitored employing on-line flow cytometry during the propagation. During the common-batch phase, no increase in fluorescence was detected (Figure 3B). In contrast, during the fed-batch phase, a clear induction of the biosensor was recorded with the H propagation once feeding was started, whereas the biosensor remained uninduced in both GX and GE propagations (Figure 3B). The robust coefficient of variance (rCV), which gives an indication of the distribution of the signal within the population, showed no increase in the H-propagated cells after induction by exposure to hydrolysate (Appendix A). Further analysis by visualization of subpopulations in histograms also showed no subpopulation, except for one of the biological replicates of H propagation, which showed a small subpopulation at the beginning of the induction (Appendix A). This small subpopulation, accounting for less than 15% of the population, corresponded mostly to permeabilized cells that were probably too damaged to exhibit a response to the sensor. 

After each propagation, the cells were collected and used for inoculation of anaerobic fermentations with increasing levels of hydrolysate, and thus increasing inhibitory strength. Four different conditions were studied corresponding to a concentration of inhibitors present in 2.5% water-insoluble solids (WIS), 5% WIS, 7.5% WIS or 10% WIS. 

After 96 h, independent of the WIS level, all fermentations reached a similar ethanol titer, around 20 g/L (Figure 4A). This indicated that yeast could tolerate inhibitor levels corresponding to those present in a 10% WIS SScF, i.e., up to 3.85 g/L of furfural, 0.34 g/L of HMF, and 2.96 g/L of acetic acid. The propagation method did not affect the final ethanol titer either. Even in the most challenging condition, 10% WISeq, high ethanol yields—ranging from 85 to 92% of the maximum theoretical yield (0.51 g/g) at the end of the fermentation (Table 2)—were obtained independently of the adopted propagation mode. However, the level of inhibitors and the propagation strategy clearly affected the *rate* of ethanol production. 

At the highest WIS content, the H-propagated cells clearly showed a faster fermentation than the cells obtained from the other propagation methods, as they started metabolizing the sugars into ethanol immediately. The GE-propagated cells were the slowest at sugar consumption, especially glucose consumption (Figure 4B), although this was not reflected in the ethanol production (Figure 4A). The faster response observed in the H-propagated cells could be translated into higher volumetric productivity, especially at the beginning of the fermentation (Table 2). 

In terms of redox response, the cells from all three propagation strategies showed an initial increase in fluorescence intensity at all WIS contents (Figure 4D). At 10% WISeq, the induction was immediate with the H-propagated cells, with the mean fluorescence reaching its peak already after 5 h; the GX-propagated cells showed a slower initial biosensor response, whereas the slowest response was observed for the GE-propagated cells (Figure 4D). At 7.5% WISeq, the three propagation strategies showed a similar fluorescence peak intensity; accordingly, a similar maximum fluorescence intensity was expected as well from all three propagation strategies at 10% WISeq. However, due to the delay in reaching the maximum response for the GX-propagated cells and an even longer delay for the GE-propagated cells, the maximum peak height for these two propagation strategies may have not been recorded because it probably occurred between the 8 and 24 h sampling times. After peaking, in all conditions, the fluorescence intensity stabilized over time with slightly higher fluorescence levels observed in the H-propagated cells. 

As the faster response of the sensor seemed to be associated with higher ethanol volumetric productivities, the fluorescence increase was plotted against the initial ethanol production rate for the highest WIS values (7.5% WISeq and 10% WISeq). Indeed, a linear correlation was observed, indicating that the initial rate of fluorescence might be an efficient online measure of the initial ethanol production rate (Figure 5).

The patterns of distribution of fluorescence in the cell population were analyzed for the biosensor response during fermentation. The highest dispersion in fluorescence intensity was observed at the condition with the levels of the inhibitors equivalent to 10% WIS, where rCV values higher than 60% were recorded for the three propagation strategies (Appendix A). Non-Gaussian-like distributions were indeed observed (Appendix A). The widest distribution of fluorescence intensity corresponded to the highest recorded fluorescence intensity at 8 h for the GX and GE-propagated cells and 5 h for the H-propagated cells. In this case, the subpopulation with a lower fluorescence did not correspond to the cells with permeabilized membranes. Thus, these results suggest that the increase in the mean fluorescence intensity observed may be due to part of the population being highly induced, whereas another part remained uninduced.

The samples were also stained with PI to determine the effect of the presence of inhibitors on the permeability of the cell membrane. A drastic decrease in the percentage of the cells that remained intact was observed over time at lower concentrations of inhibitors, i.e., 2.5% WISeq (Figure 6). In this case, after 96 h of fermentation, almost the whole cell population was permeabilized. Remarkably, no decrease was observed at higher concentrations of inhibitors, i.e., 10% WISeq, suggesting that the increase in permeabilized cells was not due to the inhibitors present in the hydrolysate but rather to the cells entering a stationary phase; indeed, the pattern correlated to the time of sugar depletion for all conditions (Figure 6, grey area). For example, at 2.5% WISeq, both glucose and xylose were already depleted after 24 h of fermentation, which is also when the decrease in the number of intact cells was first recorded. 

### 3.4. Simultaneous Saccharification and Co-Fermentation (SScF) Using the H-Propagation

To further test the yeast performance in process conditions, an SScF was performed where the whole pretreated material was instead utilized as a substrate. In this case, no biosensor was employed due to the limitations related to the presence of solids in the matrix. Based on the previously presented results, it was decided to add hydrolysate to the feed for the yeast propagation (H-propagation).Moreover, since no inhibitory effects were observed at 10% WIS-equivalent in the previous experiments, this condition was chosen as the concentration for the SScF.

After propagation, the cells were collected and used for the inoculation of SScF, together with the enzyme cocktail Cellic CTec2. During approximately the first 24 h of SScF, sugar consumption was observed and ethanol was produced, reaching a maximum ethanol yield of 41.3% for replicate A and 27.1% for replicate B. However, glucose started accumulating in the broth, suggesting that although the enzymatic hydrolysis was occurring as expected, the yeast was no longer able to consume the released glucose. No lactate production was observed, ruling out any contamination issue by lactic acid bacteria (Figure 7). 

## 4. Discussion

Monitoring and controlling the cell behavior and response to various stimuli is key to ensuring reproducible and efficient bioprocesses. Several parameters, such as temperature, pH, and O_2_ level are already routinely monitored; others, such as the level of produced CO_2,_ can also be used for feed control [25] because a high CO_2_/O_2_ ratio is an indicator of active yeast catabolism. In the present study, we investigated whether online redox monitoring by flow cytometry could be used as an additional tool to measure the cell fitness and ability to withstand environmental stressors, with a specific focus on bioethanol production from a lignocellulosic biomass. For this purpose, the previously developed TRX2p-yEGFP biosensor for redox imbalance [14] was introduced in an industrial xylose-fermenting *S. cerevisiae* strain and was evaluated under different conditions for yeast propagation and the follow-up fermentation of wheat-straw hydrolysate. The biosensor was found to be applicable to monitor the redox imbalance induced by furfural not only in a laboratory strain and defined media but also with an industrial strain and with lignocellulosic substrates. 

The direct correlation between the furfural concentration and the intensity of the fluorescence signal, previously demonstrated for furfural levels up to 3 g/L [14,24], was also observed with wheat-straw hydrolysate and the studied industrial strain. The sensor response was found to be growth-independent, with an operational limit lying over 3.8 g/L furfural for adapted cells and below 5 g/L furfural for non-adapted cells. The study also demonstrated that the signal could be recorded in complex media such as the liquid fraction of wheat-straw hydrolysate.

The introduced sensor was exploited to monitor the yeast redox fitness during the cell propagation step, an aerobic fed-batch cultivation aiming at obtaining high cell densities for further inoculation into the bioreactor for 2G ethanol production. Different feeds previously used in independent studies could then be compared to evaluate whether a correlation was found between the redox fitness during propagation and the subsequent fermentation efficiency. The baseline was set by using a mixture of glucose and xylose; for the second feed, 40% (*v*/*v*) hydrolysate was supplemented to the feed because the addition of hydrolysate during propagation has been shown to induce a short-term adaptation by pre-exposing the cells to the inhibitors present in the lignocellulosic hydrolysate [7,12]. The thirdchoice, recently proposed by Dobrescu et al. [10], used glycerol and ethanol as carbon sources as an alternative to sucrose to increase the final ethanol productivity in xylose-consuming strains. The hypothesis is that the use of non-fermentable carbon sources could stimulate a metabolic response that would enhance xylose metabolism [10]. Among the three feeding strategies, only the hydrolysate-based propagation method gave a fluorescence response, thereby strengthening the specificity of the sensor response to furfural-induced redox imbalance. Still, even under these conditions, a decrease in fluorescence was observed towards the end of the propagation, indicating that the cells were no longer under redox stress at the point of harvest. 

The sensor response was further evaluated in fermentations with the inhibitor-rich liquid hydrolysate supplemented with sugars. Independent of the propagation conditions, a redox imbalance signal was detected at the beginning of the fermentation, corresponding to the presence of furfural in all media. As expected, the signal intensity differed as a function of the level of inhibitors, but it also differed as a function of the propagation strategy, with an earlier induction of the sensor for H-propagated cells. The biosensor signal also peaked and rapidly decreased to stabilize at non-induced levels for the GX and GE propagated-cells, whereas fluorescence remained higher than the initial values with the H-propagated cells. Differences in the pattern of the sensor response have previously been attributed to the strain background [24]; however, in the present study, the same strain background was used for all experiments, indicating that the H-propagated cells displayed a long-term difference in cell behavior. It was recently shown that the transcriptome response to the presence of hydrolysate in the propagation step remains different at the time of harvest from the one obtained without added hydrolysate, despite the absence of remaining furfural in the medium; these long-lasting changes are suspected to occur via post-transcriptional regulation of transcription factors involved in the stress response, such as Msn2p, Msn4p and Yap1p [7]. This could explain the lasting difference observed in the fermentation step for the sensor response with H-propagated cells. A potential role of epigenetics in the short-term adaptation process that occurs during cell propagation can also not be ruled out [26].

The GE propagation was previously developed to obtain a higher xylose consumption rate in the following fermentation step [10], but no information was available about its impact on inhibitor tolerance. We confirmed that GE-propagated cells showed a higher xylose consumption rate than GX-propagated cells at low concentrations of inhibitors, 2.5% WISeq; however, the consumption of xylose was not faster than for H-propagated cells. With increasing concentrations of inhibitors, xylose consumption in GE-propagated cells was no longer advantageous. At 10% WISeq, GE-propagated cells showed the lowest xylose consumption rate of the three propagation strategies. Glucose consumption was also detrimentally affected in all fermentation conditions compared to H-propagated cells, which indicates that the H-propagation strategy remains the most suitable, even for xylose-fermenting strains. 

The use of on-line flow cytometry coupled with bioreactor yeast cultivations is still relatively unexplored. To the best of our knowledge, it was only applied by [27] to study the physiological properties related to the lipid accumulation in real-time in *Yarrowia lipolytica,* and Nile-red staining was used for reporting. In our approach, we introduce the possibility of using biosensors to further complement the information that can be obtained by utilizing flow cytometry. Thanks to the automatization of the sampling and rapid analysis, dynamic changes of any cellular property of interest can potentially be followed at the single-cell level and in real time with any relevant biosensor. This opens the possibility of further development of on-line flow cytometry not only as a monitoring tool but also as a controlling tool, in a similar manner as for CO_2_ control-loops [25]. Together with the recent design of compatible fluorescent proteins in *S. cerevisiae* [24,28], flow cytometry analysis also paves the way for multiple and concomitant phenotype analysis at the single- cell level.

The monitoring method was not applicable to the SScF fermentation set-up used for bioethanol production due to the presence of large particles from the solid fraction of the pretreated wheat straw that could clog the lines in the flow cytometer. In SScF, high concentrations of raw material are preferred to obtain the maximum ethanol titer and limit the water consumption and distillation costs [29]. However, the higher the WIS, the more complex the matrix due to the higher amounts of solids present and higher inhibitor concentrations in the medium. This high number of large particles is currently not compatible with the analysis of cell fluorescence [30] and clearly requires additional studies to overcome the issues. We also observed that, despite using an industrial strain that could cope up to 10% WIS equivalent in the liquid hydrolysate, the parent strain cV-110 was not able to successfully ferment the whole slurry. This indicates that the problems faced during SScF were not solely due to the presence of inhibitors. A major difference between the two set-ups was in the initial sugar composition. In the fermentations using hydrolysate, 20 g/L of glucose and 20 g/L of xylose were readily available from the start of the fermentation, whereas only xylose, which is mostly present in the hydrolysate, was readily available in the SScF.- In contrast, glucose was gradually released throughout the SScF by the action of the hydrolytic enzymes. This could indicate that a minimum glucose concentration is necessary for the cells to detoxify the inhibitors and/or assimilate the xylose in the medium. Another difference between the two set-ups was the presence of solid particles in the SScF. These particles have recently been shown to reduce tolerance to inhibitors and hamper xylose consumption in the presence of high concentrations of inhibitors [31]. Distinguishing between these possible factors will require additional investigation.

## 5. Conclusions

The TRX2p-yEGFP biosensor was shown to be able to report on redox imbalance in both defined and hydrolysate media when introduced in an industrial xylose-fermenting strain. The rapid and transient response was observed for furfural levels up to at least 3.8 g/L. The induction rate of the biosensor for redox imbalance also correlated with the ethanol production rate, showing that the biosensor can be used as a rapid indicator of the process fermentation rate. This paves the way for the use of biosensors coupled to on-line flow cytometry for rapid analysis of intracellular properties in real time during fermentations.

The study also highlighted that propagation in the presence of hydrolysate remains the most efficient strategy to obtain robust cells for fermentation of xylose-rich lignocellulosic hydrolysate from a xylose-engineered industrial strain.

## Figures and Tables

**Figure 1 jof-09-00630-f001:**
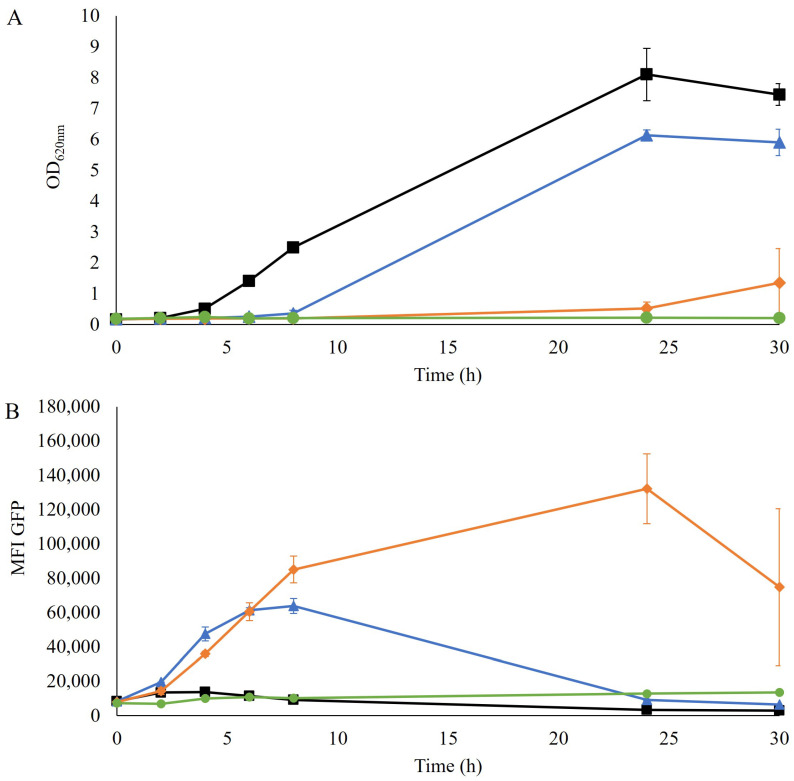
(**A**) Optical density at 620 nm and (**B**) mean fluorescence intensity (MFI) of the GFP signal over time during the aerobic cultivation of TMBRP011 in shake flasks containing YPD medium with no furfural (■), 1.25 g/L of furfural (▲), 2.5 g/L of furfural (♦) and 5 g/L of furfural (●). Biological duplicates were performed.

**Figure 2 jof-09-00630-f002:**
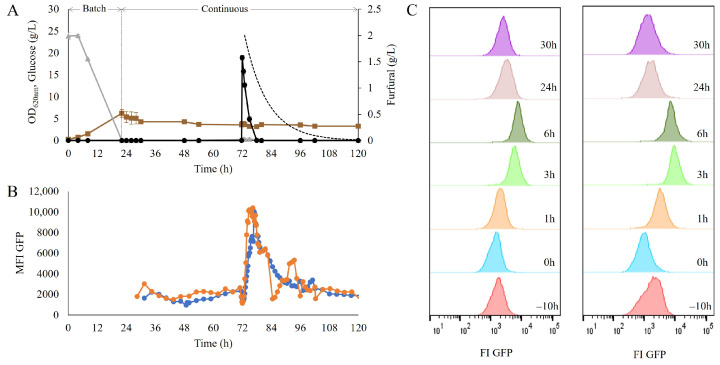
(**A**) Optical density (OD_620nm_) (■), glucose (▲) and furfural concentrations (●) over time in the chemostat cultivation of the strain TMBRP0011 in the defined mineral medium. The dotted line represents the expected furfural dilution from the bioreactor due to the continuous operation mode. (**B**) Mean fluorescence intensity on the GFP channel over time, recorded using on-line flow cytometry. As the measurements had to be recorded with slight time differences, the two biological replicates are provided separately as biological replicate 1 (●) and biological replicate 2 (●). (**C**) Histogram representation of the population distribution of the GFP fluorescence signal, recorded during the first 6 h after the addition of a pulse of 2 g/L furfural in a chemostat. Each column corresponds to one of the biological replicates.

**Figure 3 jof-09-00630-f003:**
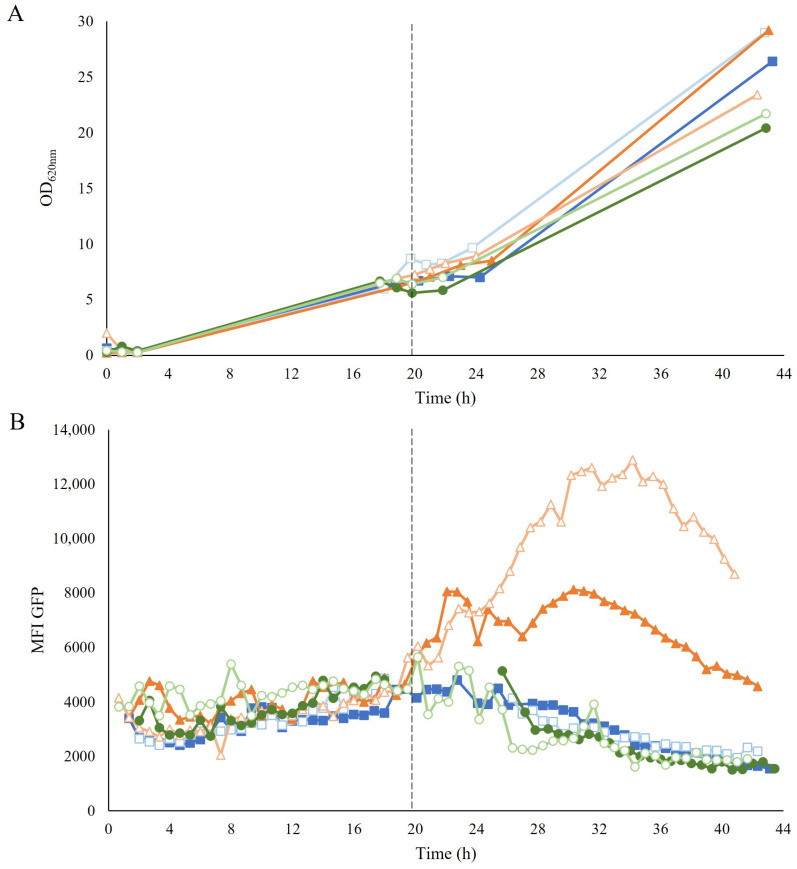
(**A**) Optical density (OD_620nm_) and (**B**) mean fluorescence intensity (MFI) of GFP over time during the aerobic fed-batch GX propagation (■), H propagation (▲), and GE propagation (●) of TMBRP011. Empty symbols correspond to the second biological replicate for each propagation strategy. The dotted line around 20 h marks the beginning of the fed-batch phase, determined by the decrease in CO_2_ production around 19–21 h after batch inoculation. Due to technical difficulties with the OnCyt equipment, it was not possible to record measurements between 19–25 h for one of the replicates.

**Figure 4 jof-09-00630-f004:**
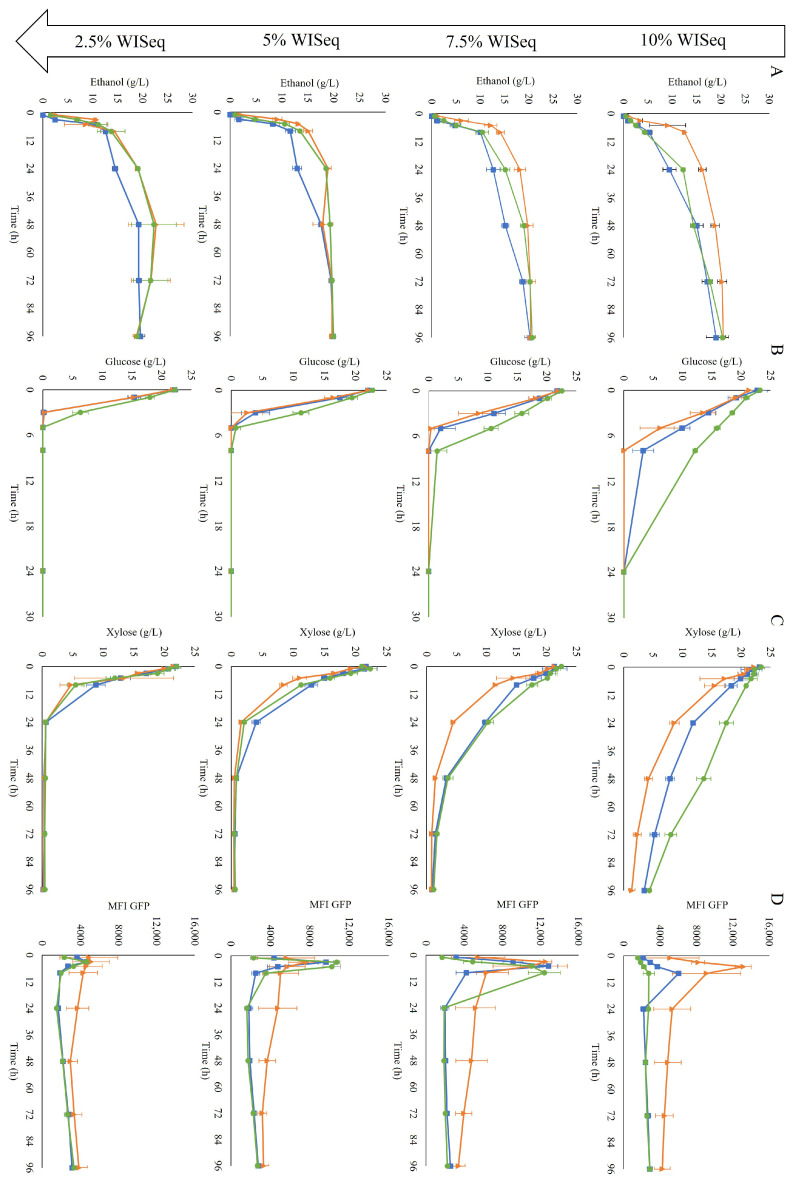
Fermentation profiles of cells collected after GX propagation (■), H propagation (▲), and GE propagation (●) in inhibitor levels corresponding to 10% WIS (first row), 7.5% WIS (second row), 5% WIS (third row), and 2.5% WIS (fourth row). Concentrations of ethanol (**A**), glucose (**B**) and xylose (**C**) over time are shown. The final column (**D**) shows the mean fluorescence intensity (MFI) of the GFP response from the biosensor for redox imbalance. Two biological replicates were performed.

**Figure 5 jof-09-00630-f005:**
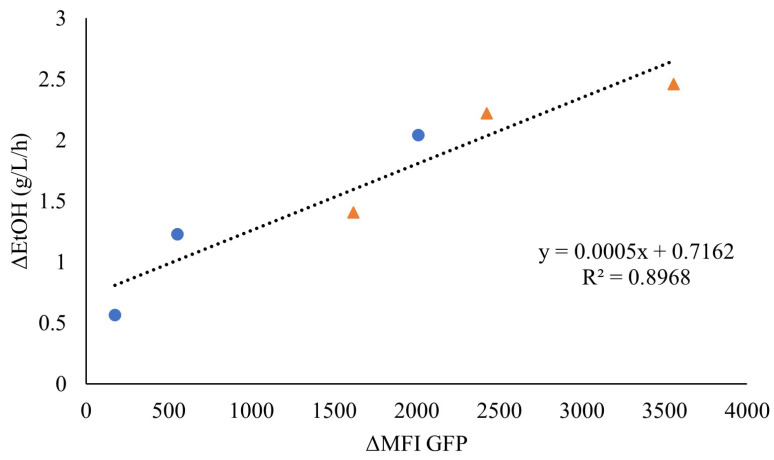
Correlation between the ethanol production rate (g/L/h) (ΔEtOH) and the rate of increase in mean fluorescence intensity (ΔMFI) until maximum fluorescence for the anaerobic cultivation of TMBRP011 in wheat-straw hydrolysate containing inhibitor levels equivalent to 10% WIS (●) and 7.5% WIS (▲).

**Figure 6 jof-09-00630-f006:**
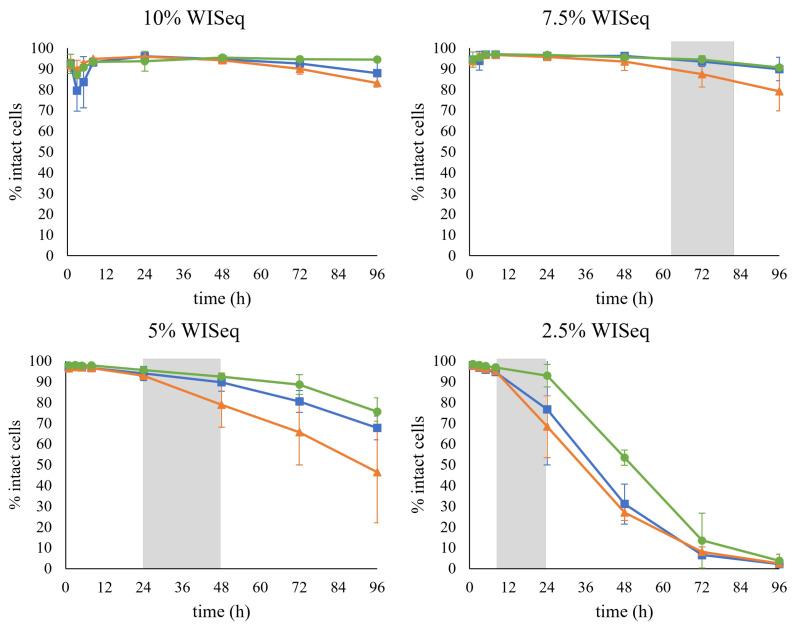
The percentage of intact cells during fermentation for the cells collected after GX propagation (■), H propagation (▲), and GE propagation (●). The gray area represents the time range at which sugar depletion was observed.

**Figure 7 jof-09-00630-f007:**
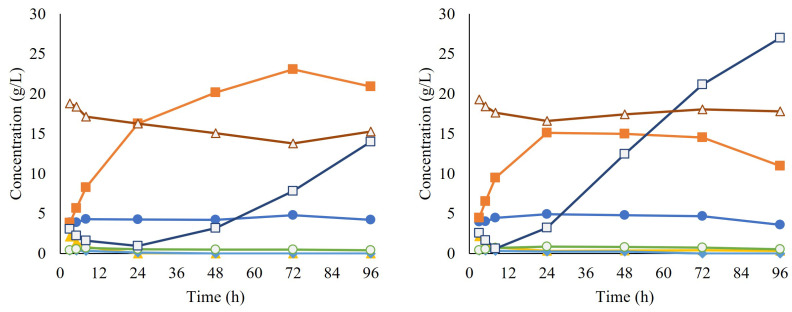
Concentrations over time of acetate (●), ethanol (■), furfural (▲), lactate (○), glucose (□), and xylose (Δ) during simultaneous saccharification and co-fermentation (SScF). Each graph corresponds to a different biological replicate.

**Table 1 jof-09-00630-t001:** Composition of the liquid fraction of the pretreated material. Sugars are presented in monomeric form.

Compound	Concentration (g/L)
Glucose	3.05
Xylose	25.69
Galactose	0.88
Arabinose	4.70
Mannose	-
Furfural	6.03
HMF	0.58
Acetic acid	4.34

**Table 2 jof-09-00630-t002:** Volumetric ethanol productivity (g/L/h) and ethanol yield as a percentage of the maximum theoretical yield obtained for anaerobic fermentations with wheat-straw hydrolysate containing 10% WIS equivalent levels of inhibitors.

Propagation Method	Volumetric Ethanol Productivity (g/L/h)					Ethanol Yield (%max)		
	8 h	24 h	48 h	72 h	96 h	48 h	72 h	96 h
GX	1.18 ± 0.07	1.89 ± 0.17	2.15 ± 0.16	2.38 ± 0.13	2.54 ± 0.25	73.7 ± 0.6	81.4 ± 2.1	86.7 ± 1.4
H	1.57 ± 0.01	2.03 ± 0.10	2.35 ± 0.11	2.54 ± 0.12	2.56 ± 0.14	84.8 ± 4.5	91.3 ± 5.0	92.3 ± 5.6
GE	0.55 ± 0.07	1.53 ± 0.05	1.80 ± 0.07	2.23 ± 0.04	2.55 ± 0.07	60.5 ± 3.2	74.9 ± 2.7	85.7 ± 1.0

## Data Availability

Not applicable.

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
