# Peer review of "Assessment of the TRX2p-yEGFP Biosensor to Monitor the Redox Response of an Industrial Xylose-Fermenting Saccharomyces cerevisiae Strain during Propagation and Fermentation"

_jof, 2023, doi:10.3390/jof9060630_

Round 1

Reviewer 1 Report

This study introduced on-line flow cytometry to assess the response of the redox biosensor TRX2p-yEGFP in an industrial xylose-fermenting strain of Saccharomyces cerevisiae during cell propagation and the following fermentation of wheat straw hydrolysate. The study highlighted that propagation in the presence of hydrolysate remains the most efficient strategy to obtain robust cells for the fermentation of xylose-rich lignocellulosic hydrolysate from a xylose-engineered industrial strain. The experimental methods and manuscripts are well-prepared, but some points need to be considered before publication.

1.     Change the term "lignocellulosic materials" to "lignocellulosic biomass" in the whole text.

2.     Keywords need to be different from the words in the title and they should begin with a capital letter.

3.     Line 32, Add the term “enzymatic” before hydrolysis.

4.     The first section of the introduction part should cite references.

5.     In the method section, how to detect the glucose and ethanol concentrations. 

Can be improve 

Author Response

  1. Change the term "lignocellulosic materials" to "lignocellulosic biomass" in the whole text.

The suggested change was made and the term lignocellulosic biomass was used instead.

  1. Keywords need to be different from the words in the title and they should begin with a capital letter.

Keywords were modified and capitalized.

  1. Line 32, Add the term “enzymatic” before hydrolysis.

The suggested change was made and the term enzymatic was added.

  1. The first section of the introduction part should cite references.

A review reference was added.

  1. In the method section, how to detect the glucose and ethanol concentrations.

A clarification for the quantification of extracellular metabolites was added in the Analytical methods section of the Materials and Methods.

Reviewer 2 Report

The manuscript “Assessment of the TRX2p-yEGFP biosensor to monitor the redox response of an industrial xylose-fermenting Saccharomyces  cerevisiae strain during propagation and fermentation” by Perruca Foncillas et al. is an interesting description to detect oxidative stress in Saccharomyces cerevisiae during industrial conditions. It is very useful to monitor this stress directly as it may reflect the metabolic performance of yeast. The authors link the levels of fluorescence with the speed of ethanol production, for instance. My main objection is that call the TRX2p-yGFP fusion a “redox sensor” could be confusing. There are fluorescent proteins that show its own redox status as change in fluorescence, being true sensors of the intracellular redox environment (10.1016/j.bbagen.2013.05.030). As a stress-dependent promotor, it reflects redox unbalances or redox response, as mentioned in the text, and show that the machinery that keeps the redox balance is been surpassed by too many ROS. The title reflects well the work, but rephrase the rest of “redox biosensor” claims.

Minor points:

Line 134: Extra “.”

L. 135. Define the integration site. Is it possible that more than one copy of the biosensor are integrated?

L. 185. Define CDW. How is it calculated?

L. 263. No induction when cells reach stationary phase? Comment growth status TRX2 expression.

L. 278. Why 2.5 g/l were not used like before?

Fig. 3. Is there a gap between 20-26h for some samples?

L. 354. Italics.

Discussion. It feels some times redundant, when results are again explained. Reduce. Indicate the limitation due to solids. Which industrial substrates could be used with this method?

Author Response

The manuscript “Assessment of the TRX2p-yEGFP biosensor to monitor the redox response of an industrial xylose-fermenting Saccharomyces  cerevisiae strain during propagation and fermentation” by Perruca Foncillas et al. is an interesting description to detect oxidative stress in Saccharomyces cerevisiae during industrial conditions. It is very useful to monitor this stress directly as it may reflect the metabolic performance of yeast. The authors link the levels of fluorescence with the speed of ethanol production, for instance. My main objection is that call the TRX2p-yGFP fusion a “redox sensor” could be confusing. There are fluorescent proteins that show its own redox status as change in fluorescence, being true sensors of the intracellular redox environment (10.1016/j.bbagen.2013.05.030). As a stress-dependent promotor, it reflects redox unbalances or redox response, as mentioned in the text, and show that the machinery that keeps the redox balance is been surpassed by too many ROS. The title reflects well the work, but rephrase the rest of “redox biosensor” claims.

Following the reviewer’s suggestion, the use of “redox sensor” has been now replaced by “biosensor for redox imbalance” where relevant to better reflect its nature.

Minor points:

Line 134: Extra “.”

The typo was corrected.

  1. 135. Define the integration site. Is it possible that more than one copy of the biosensor are integrated?

The integration site was defined in the Materials and methods. There is indeed a possibility for multiple integration in the strain that was used since it is an industrial strain with polyploidy background, but the integration should occur at the same integration locus that is defined by the XI-3 region.

  1. 185. Define CDW. How is it calculated?

The use of the abbreviation CDW as cell dry weight was incorporated in the text. Details about the methodology used for its measurement were added in the Analytical tools section of the Materials and Methods.

  1. 263. No induction when cells reach stationary phase? Comment growth status TRX2 expression.

As can be seen on Fig. 1, there was indeed no induction of the biosensor in the absence of inducer (no furfural), independently of the growth phase in which the cells were.

  1. 278. Why 2.5 g/l were not used like before?

Due to the strong growth inhibition observed when 2.5 g/L furfural were used, it was decided to lower the concentration slightly to avoid high disturbance of cell growth during the pulse which could eventually lead to a wash-out of the cells from the bioreactor while still getting a strong response from the biosensor.

Fig. 3. Is there a gap between 20-26h for some samples?

Due to technical difficulties with the on-line flow cytometry system, it was not possible to record measurements for one of the replicates during the time period between 18-25 hours of cultivation. This information has been now added in the description of Fig. 3 for clarification.  

  1. 354. Italics.

The use of italics on “rate of ethanol” was made to highlight the fact that it is the rate what is affected and not the titer.

Discussion. It feels some times redundant, when results are again explained. Reduce. Indicate the limitation due to solids. Which industrial substrates could be used with this method?

We reduced the discussion´s first part, as suggested.

The presence of solid particles with higher size than the diameter of the tubing on the flow cytometer is the limiting factor on the use of the described methodology on solid-containing configurations such as SSF. These particles could clog the lines in the flow cytometer and damage it. This clarification has been added to the updated version of the manuscript.